# Hospitalization Trends for Acute Appendicitis in Spain, 1998 to 2017

**DOI:** 10.3390/ijerph182312718

**Published:** 2021-12-02

**Authors:** Concepción Carratalá-Munuera, Jessica del Rocio Pilco, Domingo Orozco-Beltrán, Antonio Compañ, Jose A. Quesada, Rauf Nouni-García, Vicente F. Gil-Guillén, Luis García-Ortíz, Adriana López-Pineda

**Affiliations:** 1Clinical Medicine Department, School of Medicine, University Miguel Hernández de Elche, 03550 Alicante, Spain; maria.carratala@umh.es (C.C.-M.); dorozco@umh.es (D.O.-B.); jquesada@umh.es (J.A.Q.); Vte.gil@gmail.com (V.F.G.-G.); adriannalp@hotmail.com (A.L.-P.); 2Pathology and Surgery Department, Medical School, University Miguel Hernández de Elche, 03550 Alicante, Spain; jpcvvvv@gmail.com (J.d.R.P.); af.company@umh.es (A.C.); 3Salamanca Primary Care Research Unit (APISAL), Salamanca Bomedical Research Institute (IBSAL), Salamanca Primary Care Management, Castilla y León Regional Health Management (SACyL), Avenida de Portugal 83, 37005 Salamanca, Spain; lgarciao@usal.es; 4Department of Biomedical and Diagnostic Sciences, University of Salamanca, Calle Alfonso X el Sabio s/n, 37007 Salamanca, Spain

**Keywords:** acute appendicitis, regression analysis, hospital stay

## Abstract

The incidence of acute appendicitis decreased in Western countries from 1930 to at least the early 1990s, when epidemiological data started becoming scarcer. This study aimed to assess the trend in annual hospitalizations for acute appendicitis in all people Spain for a 20-year period between 1998 and 2017. This observational study analyzing direct age-standardized hospital admission rates by gender and age group (0–14 years, 15–34 years, 35–44 years, 45–64 years, and ≥65 years). Joinpoint regression models were fitted to evaluate changes in trends. There were 789,533 emergency hospital admissions for acute appendicitis between 1998 and 2017: 58.9% in boys and men and 41.1% in girls and women. Overall, there was a significant increase in admissions for this cause from 1998 to 2009, with an annual percent change (APC) of 0.6%. Following the peak in 2009, admission rates decreased by around 1.0% annually until 2017. The length of hospital stay gradually decreased from 4.5 days in 1998 to 3.4 days in 2017. The trends in hospital admissions for acute appendicitis in Spain changed over the study period, decreasing from 2009, especially in people younger than 35 years.

## 1. Introduction

The incidence of acute appendicitis decreased in high-income countries from 1930 to the early 1990s, but most published studies have not analyzed subsequent time periods [1,2]. Acute appendicitis mainly affects adolescents and young adults, with people aged 20 to 30 years carrying the highest risk, and those under 5 years the lowest [3,4,5]. In childhood, the incidence of appendicitis is higher between the ages of 8 and 15 years, increasing with age [6]. Appendicitis is especially rare in children up to the 2nd year of life, while in children between the ages of two and five it still occurs in a certain number of cases, usually with perforation/complicated appendicitis [7,8]. Over 20 years ago, the odds of suffering acute appendicitis in people aged over 50 years was estimated at 1 in 35 men and 1 in 50 women. In those older than 70 years, the odds were less than 1 in 100.

In adolescents and young adults, acute appendicitis is more frequent in men, with a ratio of 1.3 cases in men to 1 in women [9]. A recent study estimated that globally in 2019, there were 17.7 million cases of acute appendicitis (228 cases per 100,000 person-years) and more than 33,400 deaths (0.43 per 100,000 person-years) [3]. The mortality rate increases exponentially from age 30, standing at 0.2% for people under 65 years and 4.6% for older people [10]. The three main known risk factors are a poor physiological reserve in older adults, associated diseases (especially cardiorespiratory diseases), and appendicular perforation at the time of surgery [11].

Several theories have been proposed to explain this pathology. The most widely accepted among surgeons describes an initial phase characterized by obstruction of the appendicular lumen, caused by lymphoid hyperplasia, hardened fecal material, tumors, intestinal parasites, enteric or systemic bacterial infections or viral diseases, or foreign bodies like the bones of small animals or seeds. This obstruction favors the secretion of mucus and bacterial growth, generating luminal distension and increasing the intraluminal pressure. The lymphatic and venous drainage is then obstructed, further promoting bacterial growth and resulting in the production of an edema [12,13].

As far back as 1886, Reginald and Fritz [14] recommended early diagnosis and treatment to achieve a good outcome, and this maxim has not changed. In addition to the clinical interview and physical examination, the diagnosis of acute appendicitis is based on clinical and analytical data, and depending on these results, on the performance of imaging tests or even the surgical exploration of the appendix [15,16]. The most commonly recommended clinical-analytical model is the Alvarado scale, a diagnostic aid for use in patients with suspected acute appendicitis, with a sensitivity of 89.5% and a specificity of 69.2%. Recently, Appendicitis Response Inflammatory Score (AIR score [17]) has been introduced and widely used in clinical practice due to higher sensitivity and specificity than Alvarado score [18].

Since the turn of the century, new biomarkers for diagnosis have been identified, including C-reactive protein [19], while imaging tests like computed tomography (CT) have been incorporated into emergency care [20,21]. C-reactive protein, in particular, has proven highly useful for diagnosis, outperforming other biomarkers in uncomplicated acute appendicitis. CT has been shown to be helpful for confirming the diagnosis, but it has potential side effects related to ionizing radiation and contrast-induced nephropathy. Magnetic resonance imaging is an alternative to CT for the evaluation of acute appendicitis, its main advantage being the absence of exposure to ionizing radiation. In the early 1990s, the development of different types of probes and focalized compression techniques for abdominal ultrasound enabled the discrimination of acute appendicitis. However, this method has limited sensitivity (83%) and specificity (78%) compared to other diagnostic tests like the CT (94% and 90%, respectively). Moreover, its interpretation depends on the radiologist, and it is associated with some visualization problems in certain patients, for example, obese people [22,23]. On the other hand, the ultrasound also has some advantages over the CT, including greater availability in hospitals, lower cost, and lack of exposure to radiation. It therefore represents a useful technique to rule out cases with acute appendicitis, but it is not considered acceptable for cases with visualization problems, like normal appendicitis [24]. While hospitals in the USA routinely use CT as a way to avoid the costs derived from unnecessary surgery [25], in European contexts the typical protocol calls for initial use of abdominal ultrasound in case of diagnostic doubt, followed by CT in the presence of negative results or problems visualizing the appendix [26].

The appendectomy became the preferred therapy for acute appendicitis [27,28] and was classically considered the standard surgical treatment [29,30]. Nevertheless, after the German gynecologist Semm performed the first laparoscopic appendectomy in 1980, this technique was adopted both for the appendectomy itself and—in the 1990s—for diagnostic exploration of acute appendicitis, as advanced imaging tests were not yet being routinely applied [31,32]. Laparoscopy was associated with reductions in wound infections, postoperative pain, hospital stay, and costs as well as better aesthetic outcomes and an earlier return to prior activity levels [33,34,35,36]. A systematic review and meta-analysis from 2021 suggested that same-day discharge following laparoscopic appendectomy for uncomplicated appendicitis is safe and does not increase the risk of readmission, complications, or unplanned hospital visits. Moreover, it can reduce costs and improve the patient’s satisfaction [37].

In 2020, Sun Jung Oh et al. concluded that *Campylobacter jejuni* may be a major cause of acute appendicitis [38], supporting previous studies demonstrating that antibiotic prophylaxis is effective for preventing superficial infections and intraabdominal abscesses in patients with acute appendicitis. If started as soon as the patient is diagnosed with acute appendicitis, this prophylaxis may even reduce hospital admissions [39,40,41].

Some authors have postulated that acute appendicitis was the first serious pathology that emerged with the adoption of low-fiber diets. In 1969, Burkitt [42] highlighted that middle-aged people (40 to 60 years) in Uganda presented a much lower incidence of so-called Western diseases that were common in their counterparts in England and which were associated with prevalent behaviors, particularly dietary patterns, in high-income countries. This author concluded that the diseases linked to these behaviors represented a threat to public health in high-income settings, and it was necessary to raise the public’s and government’s awareness of the need to increase intake of foods rich in fiber. Moreover, current evidence suggests that these behavioral changes could also increase quality of life by reducing the effects of associated diseases [43]. The primary objective of the present study was to evaluate the temporal trends of hospital admissions for acute appendicitis in Spain in the period 1998–2017. The secondary aims were to assess whether there are variations in these trends by gender and age, as well as to analyze the evolution of the median length of hospital stay.

## 2. Materials and Methods

This observational study of temporal trends at a national level studied the population residing in Spain from 1998 to 2017. We included people of any age with an emergency hospital admission for acute appendicitis in Spain during the 20-year study period. Exclusion criteria were missing data or unknown values for the study variables or hospital stays of 12 days or more (to eliminate outliers).

Hospital admissions data came from the minimum basic data set (a unique database for Spain) [44]. This set of information is collected at hospital discharge and contains health and administrative information from all hospitals of the Spanish Health System. This database began to include data from private hospitals progressively in 2004. The diagnosis on admission was defined according to the Spanish version of the 9th and 10th revisions of the International Classification of Diseases (ICD), using the diagnostic code ICD-9-MC 540 for the period from 1998 to 2015 and ICD-10-ES for 2016 and 2017. The source for population data was the continuous civil registry, maintained by the National Statistics Institute [45]. Study variables were gender, age in years, year of admission (1998 to 2017), and length of hospital stay in days.

## 3. Statistical Analysis

Categorical variables were described as frequencies and quantitative variables as means (standard deviation (SD)) and ranges. We used direct methods to calculate the age-standardized rates (ASRs) of hospital admissions and their 95% confidence intervals (CIs) according to the 2013 standard European population. We calculated the admission rates for all age groups (0–14 years, 15–34 years, 35–44 years, 45–64 years, and ≥65 years) for each year from 1998 to 2017, stratifying results by gender. The SPSS statistical package (v.26) was used for the descriptive analysis.

To analyze the temporal trends in hospital admissions and detect any significant changes during the study period, we fitted joinpoint regression models for the overall population and by age group and gender. These models provided an estimate for the annual percent change (APC) in the ASR for each identified segment. A negative APC indicates a downward trend, and vice versa. The models were fitted under the assumption of autocorrelation errors in the data, and they were selected using the permutation test, with a minimum of 0 joinpoints and a maximum of 3. The Joinpoint Regression Program (v.4.6.0) of the US National Cancer Institutes was used for the analysis [43]. The ASRs were represented graphically for each age group and gender, together with the estimated joinpoint segments. Finally, the temporal variation in the length of stay was analyzed according to the median days of hospital stay and the and interquartile range due to acute appendicitis, by sex and age group for each age group and gender, with graphic visualization of the results [46].

## 4. Results

There were 789,533 emergency admissions for acute appendicitis from 1998 to 2017, 58.9% in male and 41.1% in female. By age group, admissions were distributed as follows: 0–14 years, 25.3%; 15–34 years, 41.1%; 35–44 years, 12.2%; 45–64 years, 13.8%; and 65 years or older, 7.6% (Table 1). Patients’ mean age was 29.8 years (range 0 to 111). Appendix A shows the ASR for admissions in each year of the study period by gender and age group.

According to the joinpoint regression (Figure 1), admissions due to acute appendicitis showed a significant upward trend in male until 2009, with an APC of 0.6%. Following this peak, the direction of the trend changed, decreasing by an average 1.0% annually. By age group, admissions peaked in 2013 in boys aged 0–14 years (Appendix A), in 2009 in those aged 15–34 years (Appendix A), and in 2005 in those aged 35–44 years (Appendix A), subsequently rates started to decline. On the other hand, an upward trend was apparent in those aged 45 years or older (Appendix A), especially in patients aged 65 years or more from 2006 (Appendix A).

A comparable pattern is apparent in female (Figure 2), with admissions peaking in 2009, followed by a significant decline (average APC 0.7%). By age group, this decline began from 2010 in girls aged 0–14 years (Appendix A), from 2009 in those aged 15–34 years (Appendix A), and from 2010 in female aged 35–44 years (Appendix A). As with male, this trend is different in female older than 45 years, where from 2006, data show an APC of 1.5% in the 45–64-year age group (Appendix A) and of 1.4% in those aged 65 or older (Appendix A).

Appendix A shows the median length of stay and interquartile range for each included study year by gender and age group. There was a gradual decline in length of stay for acute appendicitis over the study period, from 4 days in 1998 to 3 days in 2017, with similar trends across genders (Figure 3) and age groups, although the median length of stay was longer in older people. 

## 5. Discussion

Our analysis showed that hospital admission trends for acute appendicitis in Spain changed between 1998 and 2017, with similar patterns observed in male and female. Following a steady increase in annual admission rates from 1998 to 2009, the trend then reversed course, with a subsequent decline in admissions. This pattern differed in people aged 45 years or older, and especially in those older than 65 years, in whom admission rates started to climb from 2006. In people younger than 45 years, admission rates peaked in 2009 and then began a slightly irregular descent that lasted until 2016 and was more pronounced in female compared to male. In adults aged 45 to 64 years, the trend shifted upward in 2009 and remained on this course to 2017, while in those aged 65 or older, admission rates increased until 2016 before a discreet drop in 2017. The length of stay showed a downward trend across ages and genders and throughout the study period.

Over the past years, studies on the incidence of acute appendicitis and/or appendectomies present contradictory and heterogeneous results. Ferries et al. [4] undertook a meta-analysis in 2017, comparing the incidence and temporal trends of appendicitis and appendectomies worldwide. The review authors concluded that incidence was stable in most Western countries, and they reported a pooled incidence for acute appendicitis and appendectomy in Spain that is consistent with our findings. In countries with more recently industrialized economies, such as South Korea, Turkey, and Chile, the scarce data available suggest that appendicitis is increasing rapidly [4]. Another study, performed in Norway [47], prospectively analyzed the incidence of acute appendicitis from 1989 to 1998, finding stable rates for the 10-year study period. In Finland, Ivles et al. [48] reported that the incidence of acute appendicitis and appendectomies decreased from 1987 to 2007; however, in the USA, Coward et al. [49] reported an upward trend from 2000 to 2008.

In Spain, the National Health Technology Assessment Agency, working under the Ministry of Health and Consumption [50], suggested that it would be useful to introduce ultrasounds in primary health care. Some Spanish scientific societies then began including practical training in ultrasounds among their objectives [51,52]. A few regional health systems have decisively moved to equip health centers with ultrasound equipment and train professionals in their use, including the Community of Madrid (since 2009), Galicia, Catalonia, and the Balearic Islands [53,54]. 

In Spain, the current use of ultrasound by non-radiologists, a practice known as point-of-care ultrasound, bedside ultrasound, or clinical ultrasound, is meant to make this technique accessible in the place and time of the clinical consultation [54]. Over the last two decades, the progressive implementation of the abdominal ultrasound in primary health care [53] would have favored reduced hospital referrals to reach an adequate differential diagnosis, while access to CTs in hospital emergency services [19] could have reduced the frequency of false positives. Our results support this hypothesis, as we observed a decrease in hospital admissions from 2009. Similarly, the results of a study performed in the UK in 2020 showed that the greater use of the CT in the context of the COVID-19 pandemic enabled the identification of simple appendicitis for conservative treatment and reduced the negative appendectomy rate [55].

Previous studies have shown that laparoscopic treatment for appendicitis is associated with a reduced length of hospital stay in adults [36,56], in keeping with our results. Several studies [5,57,58] that analyzed variations in the incidence of appendicitis between and within countries, ethnic groups and professionals suggest that rates are highest in the regions where the diet is characterized by low fruit and vegetable intake and high consumption of processed and low-fiber foods. In contrast, the pathology is rare in countries consuming traditional, unprocessed foods with more fiber [4]. The Mediterranean diet, followed widely in Spain, consists of abundant intake of fruits, vegetables, legumes, nuts, whole grains, and olive oil; moderate consumption of fish, poultry, pork, and low-fat dairy; and low intake of red meat and processed foods [59]. The differences in fiber intake could be one reason for the geographical variations in the incidence of appendicitis within a single country [5,57,58].

The trends observed in this study may be due to incremental improvements in the diagnosis and pharmacological treatments since the turn of the century [19,39,40,41,53]. It is feasible that acute appendicitis has been underdiagnosed in people aged 45 years or older due to the higher incidence in adults under the age of 30 [60].

These somewhat contradictory trends may be due to the interplay between diverse geographical, educational, environmental, genetic, or diagnostic factors. Some patients with non-specific abdominal pain may be treated conservatively, avoiding the manifestation of acute appendicitis. Likewise, an erroneous diagnosis of suspected infection in another site in the abdominal cavity, when treated with antibiotics, can help decrease the rate of admissions for acute appendicitis. Another relevant factor is the use of an administrative registry of hospital discharges versus a registry of pathological anatomy reports. The former source will generally result in an overestimation of the number of patients with acute appendicitis.

One strength of our study was the use of public databases as information sources for this study, as these official sources are subject to quality control and provide clean, revised data on hospital admissions in Spain, with few data management errors [4]. We were unable to establish associations between admissions trends and patient characteristics that could explain the temporal changes detected, as the admissions database does not provide these data.

The results of this study show a change in trends that could be influenced by diverse modifiable factors. Future studies could also explore associations with individual variables that might explain the temporal changes detected.

## 6. Conclusions

Our analysis reveals a change in trends of hospital admissions for acute appendicitis in Spain between 1998 and 2017, with a similar pattern in male and female. Admission rates fell from 2009, probably due to gradual advances in diagnostic tests, such as the ultrasound, CT, and antibiotic treatment. By age groups, admissions started to descend among people under 35 years of age in 2009, with no difference by gender; however, this trend was not observed in middle-aged people, and rates actually increased in older people. Median length of stay decreased over the study period, both overall and in all age groups and genders, probably due to the adoption of laparoscopic appendectomy as a standard surgical technique.

## Figures and Tables

**Figure 1 ijerph-18-12718-f001:**
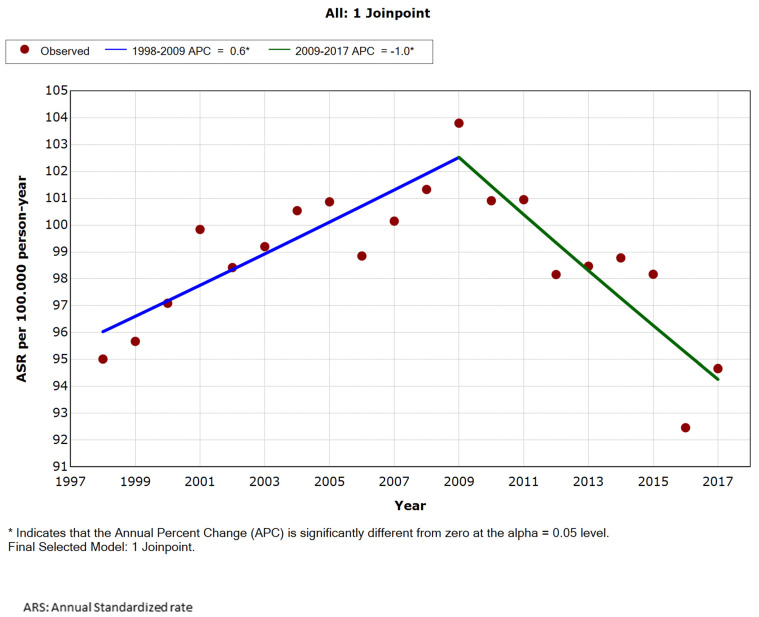
Annual age-standardized rate per 100,000 male (all ages), admitted for acute appendicitis in Spain, 1998 to 2017.

**Figure 2 ijerph-18-12718-f002:**
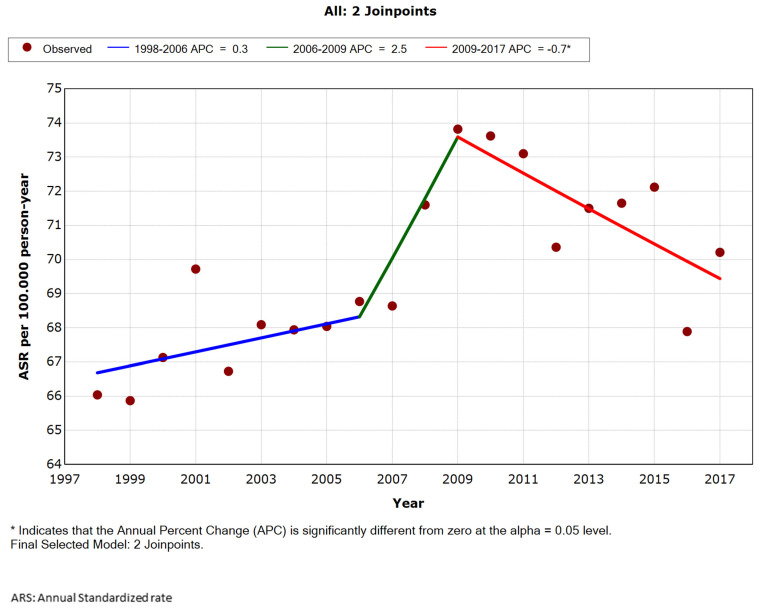
Annual age-standardized rate per 100,000 female (all ages), admitted for acute appendicitis in Spain, 1998 to 2017.

**Figure 3 ijerph-18-12718-f003:**
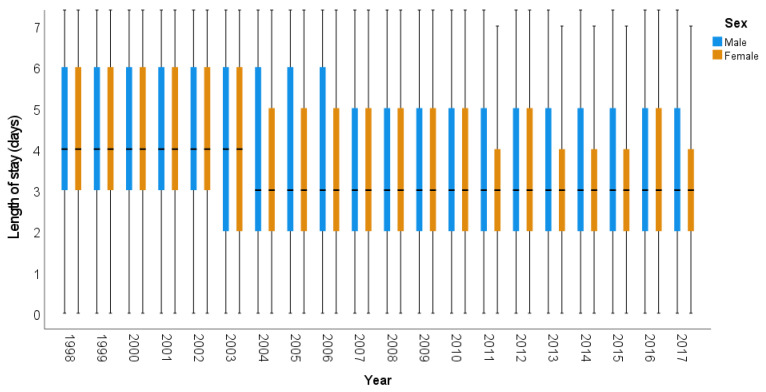
Box diagram for length of stay during admission for acute appendicitis, by year and gender.

**Table 1 ijerph-18-12718-t001:** Trend segments and annual percent change (APC) in age-standardized hospital admission rates for acute appendicitis in Spain, obtained using joinpoint regression models, by sex and age group, 1998 to 2017.

Gender	Age Group	Time Segment	APC	*p*-Value	95% CI	AAPC	95% CI	*p*-Value
Male	All	1998–2009	0.6	<0.001	(0.4, 0.8)	−0.1	(−0.3, 0.1)	0.340
		2009–2017	−1.0	<0.001	(−1.4, −0.7)			
	0–14 years	1998–2013	0.1	0.399	(−0.1, 0.4)	−0.7	(−1.2, −0.3)	0.001
		2013–2017	−3.8	0.001	(−5.8, −1.8)			
	15–34 years	1998–2003	2.1	<0.001	(1.6, 2.6)	0.0	(−0.5, 0.5)	0.985
		2003–2009	0.4	0.059	(−0.0, 0.9)			
		2009–2012	−2.4	0.134	(−5.6, 0.9)			
		2012–2017	−1.1	0.001	(−1.6, −0.6)			
	35–44 years	1998–2005	1.5	0.001	(0.7, 2.2)	0.5	(0.2, 0.8)	0.001
		2005–2017	−0.1	0.440	(−0.4, 0.2)			
	45–64 years	1998–2017	0.3	0.004	(0.1, 0.6)	0.3	(0.1, 0.6)	0.004
	≥65 years	1998–2006	−0.9	0.157	(−2.2, 0.4)	0.2	(−0.4, 0.8)	0.533
		2006–2017	1.0	0.010	(0.3, 1.7)			
Female	All	1998–2006	0.3	0.176	(−0.2, 0.8)	0.2	(−0.7, 1.1)	0.640
		2006–2009	2.5	0.391	(−3.5, 8.9)			
		2009–2017	−0.7	0.003	(−1.2, −0.3)			
	0–14 years	1998–2010	−0.1	0.771	(−0.5, 0.4)	−1.1	(−1.5, −0.7)	<0.001
		2010–2017	−2.8	<0.001	(−3.8, −1.8)			
	15–34 years	1998–2006	0.8	0.001	(0.4, 1.1)	0.4	(−0.5, 1.2)	0.403
		2006–2009	3.3	0.246	(−2.5, 9.5)			
		2009–2017	−1.1	<0.001	(−1.5, −0.7)			
		1998–2010	1.8	<0.001	(1.2, 2.3)	1.2	(0.7, 1.7)	<0.001
	35–44 years	2010–2017	0.3	0.561	(−0.8, 1.4)			
	45–64 years	1998–2017	1.5	<0.001	(1.3, 1.7)	1.5	(1.3, 1.7)	<0.001
	≥65 years	1998–2006	−1.0	0.086	(−2.2, 0.2)	0.4	(−0.2, 1.0)	0.179
		2006–2017	1.4	<0.001	(0.8, 2.1)			

APC: annual percent change; AAPC: average annual percent change; CI: confidence interval.

## Data Availability

All the data used for this analysis can be confirmed at any time.

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
