# Peer review of "Hospitalization Trends for Acute Appendicitis in Spain, 1998 to 2017"

_ijerph, 2021, doi:10.3390/ijerph182312718_

Round 1
Reviewer 1 Report
The paper is dealing with an interesting and complicated subject. The paper is well organized and clear in its structure. Being a study based on statistical data and no clinical reports, some aspects which could be interesting are not touched and veryfied. Appendicitis and appendectomy are underlying many different factors. Dealing with a large number of patients and different age groups, it is difficult for the authors to enter in detailed questions. For exmple, there is no information about conservative treatment which might have an incidence on the total numbers in the last years. Nevertheless this study provides interesting data and some stimulating upcoming questions. It could be a basis for a further analysis dealing with some more detailed problems.
I would consider this paper to be accepted for publication after minor review of the english manuscript.
Author Response
Dear Reviewer,
We appreciate all the contributions you have made in your review and that have helped to improve the quality of this article. We have taken all your comments into account and made whatever changes were possible. Modifications have been made with the track changes. Below we answer the questions raised.
Reviewer 1
The paper is dealing with an interesting and complicated subject. The paper is well organized and clear in its structure. Being a study based on statistical data and no clinical reports, some aspects which could be interesting are not touched and veryfied. Appendicitis and appendectomy are underlying many different factors. Dealing with a large number of patients and different age groups, it is difficult for the authors to enter in detailed questions. For exmple, there is no information about conservative treatment which might have an incidence on the total numbers in the last years. Nevertheless this study provides interesting data and some stimulating upcoming questions. It could be a basis for a further analysis dealing with some more detailed problems.
I would consider this paper to be accepted for publication after minor review of the english manuscript.
>> Thank you very much for your comments. We agree with your comment about conservative treatment but, unfortunately, we do not have data of this. We consider their possible relevance and the influence in recent years in the introduction of the manuscript (lines 117-122).
Reviewer 2 Report
The authors investigated the hospitalization trends for acute appendicitis in Spain from 1998 to 2017.
My remarks:
- The authors used a civil registry, maintained by the National Statistics Institute. My question: is this registry complete and accurate, and is there any other in Spain or is this the only one? Furthermore included were patients who underwent appendectomy (acute appendicitis) but there is no information on which percent of them had appendicitis. What was the rate and percentage of negative appendectomies?
- Authors excluded appendectomies with hospital stay more than 12 days...? Why is this? Bias?
- Furthermore, there is no division from simple to advanced and complicated appendicitis. It would be very interesting to see the trends between these two groups because most of the cases of simple appendicitis nowadays are discharged on the same day of surgery.
- We are at the end of 2021, why authors haven't included these last 3-4 years?
- Introduction - The authors stated that acute appendicitis mainly affects adolescents and young adults, with people aged 20 to 30 years carrying the highest risk, and those under 5 years the lowest. The incidence of acute appendicitis is also known to be high in childhood, foremost at 8 and 14 years of age. This needs to be mentioned in the introduction. Appendicitis is especially rare in children up to the 2nd year of life, while in children between the ages of two and five it still occurs in a certain number of cases, usually with perforation / complicated appendicitis (Add a reference).
- REFERENCE: authors mentioned AIR scoring but made reference only to Alvarado, and there is no reference for AIR score!
- Methodology – Clearly state primary and secondary outcomes!
- Results – It would be more appropriate to present median age than presented mean age, with interquartile range.
- Results – it is more appropriate to use the terms ‘’Male and Female’’ instead of ‘’Men and Women’’, moreover, the children were included. Also, exact p values for each group should be stated instead of generally mentioned p<0.05.
- Figure 4 – Length of stay should be presented in medians, not means. It is impractical to use shades of the same color for lines in the diagram. More different colors would be more reader-friendly.
Author Response
We appreciate all the contributions you have made in your review and that have helped to improve the quality of this article. We have taken all your comments into account and made whatever changes were possible. Modifications have been made with the track changes. Below we answer the questions raised.
1-The authors used a civil registry, maintained by the National Statistics Institute. My question: is this registry complete and accurate, and is there any other in Spain or is this the only one? Furthermore included were patients who underwent appendectomy (acute appendicitis) but there is no information on which percent of them had appendicitis. What was the rate and percentage of negative appendectomies?
>> Thank you for your comment. The registry used in this analysis of trends is unique and of the entire Spanish state. This registry refers to the information at discharge, so the patients included in this study were confirmed acute appendicitis. We have clarified this information in lines 147-150.
2- Authors excluded appendectomies with hospital stay more than 12 days...? Why is this? Bias?
>> Hospital stays longer than 12 days were excluded as they were considered relatively extreme values that could affect the estimation of average stays. Attending the reviewer comment, all length stays have been included, estimating the median and the interquartile range. Supplementary table 2 and figure 3 have been modified, where box diagrams are now shown with medians and interquartile ranges for each year, separated by sex. Figure 4 showing the mean length of stay by gender and age group has been removed since it is not feasible to calculate a box diagram for each age group in the same figure, and these medians values are also shown in full in supplementary table 2. Thank you for your comment.
3-Furthermore, there is no division from simple to advanced and complicated appendicitis. It would be very interesting to see the trends between these two groups because most of the cases of simple appendicitis nowadays are discharged on the same day of surgery.
>> Our study database includes the diagnosis at discharge and data about complications of appendicitis were not requested because this study was designed only to evaluate trends in hospital admissions for acute appendicitis, not for their evolution. Thanks for your comment, it will be taken into account for later studies.
4- We are at the end of 2021, why authors haven't included these last 3-4 years?
>> This study was carried out in 2019 and the data of 2018 year were not available on that date. We suffered a one-year delay in the publication process due to the Covid- 19 pandemia. The process of requesting data until they are sent to us takes several months, which is why we are not able to include new data in this version of the manuscript.
5- Introduction - The authors stated that acute appendicitis mainly affects adolescents and young adults, with people aged 20 to 30 years carrying the highest risk, and those under 5 years the lowest. The incidence of acute appendicitis is also known to be high in childhood, foremost at 8 and 14 years of age. This needs to be mentioned in the introduction. Appendicitis is especially rare in children up to the 2nd year of life, while in children between the ages of two and five it still occurs in a certain number of cases, usually with perforation / complicated appendicitis (Add a reference).
>> Thank very much for your suggestion. We have added this information in the lines 48-52 and we have included the following references.
Rautava L, Rautava P, Sipilä J, Kytö V. Occurrence and Treatment of Pediatric Appendicitis in Finland 2004-2014. J Surg Res. 2018 Dec;232:33-38. doi: 10.1016/j.jss.2018.06.010. Epub 2018 Jun 30. PMID: 30463737.
Zavras N, Vaos G. Management of complicated acute appendicitis in children: Still an existing controversy. World J Gastroin-test Surg. 2020 Apr 27;12(4):129-137. doi: 10.4240/wjgs.v12.i4.129. PMID: 32426092; PMCID: PMC7215970.
Pogorelić Z, Domjanović J, Jukić M, Poklepović Peričić T. Acute Appendicitis in Children Younger than Five Years of Age: Diagnostic Challenge for Pediatric Surgeons. Surg Infect (Larchmt). 2020 Apr;21(3):239-245. doi: 10.1089/sur.2019.175. Epub 2019 Oct 16. PMID: 31618143.
6- REFERENCE: authors mentioned AIR scoring but made reference only to Alvarado, and there is no reference for AIR score!
>> Thank you for your comment. In the new version of the manuscript we have included the reference for AIR score (reference number 17): Jose T, Rajesh PS. Appendicitis Inflammatory Response Score in Comparison to Alvarado Score in Acute Appendicitis. Surg J (N Y). 2021 Jul 19;7(3):e127-e131. doi: 10.1055/s-0041-1731446. PMID: 34295969; PMCID: PMC8289675.
7- Methodology – Clearly state primary and secondary outcomes!
>> We have clarified all study objectives at the end of the introduction. Please, see lines 133-137. Thank you for your suggestion.
8- Results – It would be more appropriate to present median age than presented mean age, with interquartile range.
>> We have modified the analysis of the length hospital stay. Supplementary table 2 and figure 3 have been modified, where box diagrams are now shown with medians and interquartile ranges for each year, separated by sex. Figure 4 showing the mean length of stay by gender and age group has been removed since it is not feasible to calculate a box diagram for each age group in the same figure, and these medians values are also shown in full in supplementary table 2. Thank you for your comment.
9- Results – it is more appropriate to use the terms ‘’Male and Female’’ instead of ‘’Men and Women’’, moreover, the children were included. Also, exact p values for each group should be stated instead of generally mentioned p<0.05.
>> We agree with reviewer comment and we have changed these terms. In addition, the exact p-value with 3 decimal places has been added in table 1. The mention * p <0.05 has been eliminated.
10- Figure 4 – Length of stay should be presented in medians, not means. It is impractical to use shades of the same color for lines in the diagram. More different colors would be more reader-friendly.
>> Supplementary table 2 and figure 3 have been modified, where box diagrams are now shown with medians and interquartile ranges for each year, separated by sex. Figure 4 showing the mean length of stay by gender and age group has been removed since it is not feasible to calculate a box diagram for each age group in the same figure, and these medians values are also shown in full in supplementary table 2. Thank you for your comment.
Reviewer 3 Report
This paper summarizes a huge amount of data on hospital admissions for acute appendicitis in a large population in the industrialized Europe. Due to a high quality of the medical service to their citizens for a long time the data have been prospectively collected and thereby providing rather reliable information.
The paper is fluently written and the findings and the information in the paper is of interest for the medical as well as the administrative population. This study can be repeated.
- I noted that the authors did not comment on the use of Magnetic Resonance Imaging in the diagnosis of appendicitis.
- I miss any information on excluded hospital or institutions that may confound the data.
- I don’t know if Spain has private hospitals taking care of emergencies as acute appendicitis and if so is the case, are the report from private hospitals available for research?
This amount of data reported in the paper will serve as a benchmark for other parts of the world.
Author Response
Dear reviewer,
We appreciate all the contributions you have made in your review and that have helped to improve the quality of this article. We have taken all your comments into account and made whatever changes were possible. Modifications have been made with the track changes. Below we answer the questions raised
1-I noted that the authors did not comment on the use of Magnetic Resonance Imaging in the diagnosis of apendicitis:
>> Thanks for your input. We have added the following sentence in the lines 88, 89, 90 of the manuscript: “Magnetic resonance imaging is an alternative to CT for the evaluation of acute appendicitis, its main advantage being the absence of exposure to ionizing radiation”.
2-I miss any information on excluded hospital or institutions that may confound the data.
3-I don’t know if Spain has private hospitals taking care of emergencies as acute appendicitis and if so is the case, are the report from private hospitals available for research?
>> The registry of hospital discharges began with the approval of the Minimum Basic Data Set in 1987 by the Interterritorial Council for all hospitals of the National Health System. Starting in 2004, the Minimum Basic Data Set began a process of expanding borders, gradually expanding to the private sector and incorporating data on alternative healthcare modalities to hospitalization. In 2018, 85% of hospital discharges correspond to the hospitals of the National Health System. We have clarified this issue in lines 149-150 of the manuscript. Thank you for your comments.
Round 2
Reviewer 2 Report
Authors did what was asked from them.
Congratulations.
Best regards!